# Mind the Privacy Unit! User-Level Differential Privacy for Language Model Fine-Tuning

**Lynn Chua**
Google Research
chualynn@google.com

**Badih Ghazi**
Google Research
badihghazi@gmail.com

**Yangsibo Huang**
Princeton University
yangsibo@google.com

**Pritish Kamath**
Google Research
pritishk@google.com

**Ravi Kumar**
Google Research
ravi.k53@gmail.com

**Daogao Liu**
University of Washington
dgliu@uw.edu

**Pasin Manurangsi**
Google Research
pasin@google.com

**Amer Sinha**
Google Research
amersinha@google.com

**Chiyuan Zhang**
Google Research
chiyuan@google.com

## Abstract

Large language models (LLMs) have emerged as powerful tools for tackling complex tasks across diverse domains, but they also raise privacy concerns when fine-tuned on sensitive data due to potential memorization. While differential privacy (DP) offers a promising solution by ensuring models are "almost indistinguishable" with or without any particular privacy unit, current evaluations on LLMs mostly treat each example (text record) as the privacy unit. This leads to uneven user privacy guarantees when contributions per user vary. We therefore study user-level DP motivated by applications where it is necessary to ensure uniform privacy protection across users. We present a systematic evaluation of user-level DP for LLM fine-tuning on natural language generation tasks. Focusing on two mechanisms for achieving user-level DP guarantees, Group Privacy and User-wise DP-SGD, we investigate design choices like data selection strategies and parameter tuning for the best privacy-utility tradeoff.

## 1 Introduction

Modern large language models (LLMs) are shown to be capable of solving a wide range of complex understanding and reasoning tasks (Achiam et al., 2023; Anil et al., 2023; Bubeck et al., 2023). Fueling their adoption, these pre-trained models can be further fine-tuned for specialized domains such as legal (Cui et al., 2023; Guha et al., 2023), medical (Singhal et al., 2022; Luo et al., 2022; Moor et al., 2023; Lee et al., 2023), and coding assistance (Chen et al., 2021; Li et al., 2023a; Roziere et al., 2023; Guo et al., 2024). However, this proliferation of LLMs has introduced privacy concerns: Fine-tuning on domain data risks unintentionally retaining sensitive information, which could potentially be extracted later (Carlini et al., 2021). Moreover, the trend towards using larger model sizes with more parameters (Kaplan et al., 2020) amplifies memorization risks (Carlini et al., 2022).

To address these privacy challenges, a growing body of work has focused on fine-tuning LLMs with differential privacy (DP) guarantees, a widely adopted notion to provide privacy guarantees in machine learning pipelines. At a high level, DP guarantees that a model trained with and without any particular data point is "almost indistinguishable". A training algorithm satisfying DP would ignore sensitive information of individual records but can still learn the distributional properties of the dataset. Typical DP training algorithms (e.g., DP-SGD, Abadi et al., 2016) provide such guarantees by limiting the contribution of each record to the trained model and adding calibrated noise.

Existing studies have explored DP fine-tuning techniques for various applications, such as natural language understanding (Li et al., 2022; Yu et al., 2021; 2022), generation (Yu et al., 2022; Anil et al., 2022; Yue et al., 2023), and in-context learning (Tang et al., 2024; Wu et al.,

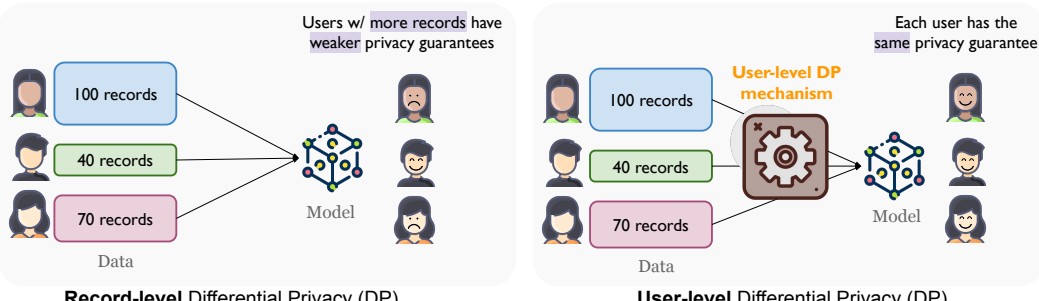

Figure 1: The privacy unit is a crucial parameter for DP guarantees. Previous works commonly treat each training example or record as the unit of privacy (i.e., record-level DP, shown on the left). However, this leads to unequal privacy protection when users contribute varying numbers of records — users with more records unfortunately obtain weaker privacy guarantees under record-level DP. To address this discrepancy, we consider the stronger requirement of user-level DP (shown on the right) for use cases where it might be necessary or applicable. User-level DP ensures uniform privacy protection across all users, regardless of each individual's number of contributed records.

2023; Duan et al., 2023). Many training algorithms and evaluations in the literature make the (implicit) assumption that each user only contributes a single record to the training dataset. This notion is usually referred to as Example-level DP or Record-level DP. In the context of language models, a record is a *sequence* of (tokenized) text. However, in many real-world scenarios, each user may provide multiple records[1]. Under Example-level DP, users contributing more records inevitably obtain weaker privacy protection — an undesirable outcome (see Figure 1). To ensure uniform privacy protection regardless of the number of records per user, we consider the stronger requirement of User-level DP, which ensures that each user obtains the same privacy guarantee. This shift from Example-level DP to User-level DP forms the core focus of our study.

We present a systematic empirical evaluation of User-level DP for fine-tuning LLMs on natural language generation tasks. Unlike existing relevant studies that mostly focus on the setting of (private) federated learning (McMahan et al., 2018; Kairouz et al., 2021; Xu et al., 2023), we do not assume a distributed or federated setting. Instead, we consider the scenario where the language modeling task[2] involves data from multiple users, and it is desirable to guarantee privacy at the user level, rather than the traditional record level. We identify and evaluate the two mainstream mechanisms for achieving user-level DP: Group Privacy and User-wise DP-SGD, as described in Section 2. We systematically propose and evaluate data selection strategies for both methods (Section 4 and Section 5) to improve the privacy-utility trade-off and compare their performance (Section 6). We hope the findings provide valuable empirical references for practitioners working on User-level DP for language modeling tasks.

Furthermore, we provide a case study of applying the algorithm from Asi & Liu (2024) to our language model fine-tuning tasks. While many empirical studies focused on Example-level DP, the theory community has studied User-level DP extensively in the past few years, with a long line of algorithms with provable guarantees. Among those, Asi & Liu (2024) currently achieve the state-of-the-art excess rate (see Section 7 for more details). However, the theoretical results typically rely on assumptions such as that the data being drawn i.i.d. from an underlying distribution, and it has not been systematically evaluated whether those assumptions hold in practice. In this paper, we present the first such study, by focusing on the algorithm from Asi & Liu (2024), which has the theoretically optimal excess rate. Our results show that the assumptions could be too restrictive for real world User-level DP applications, and call for the research community to consider more realistic assumptions.

We note that an independent and concurrent work of Charles et al. (2024) also studies user-level DP for LLM fine-tuning. While their key observations are largely consistent with ours, we discuss some differences in Section 7.

---

[1]For instance, patients contributing various medical records to train healthcare AI assistants, or users submitting different prompts/conversations to an open-domain chatbot.

[2]We focus on natural language generation tasks, which underpin many key LLM applications.

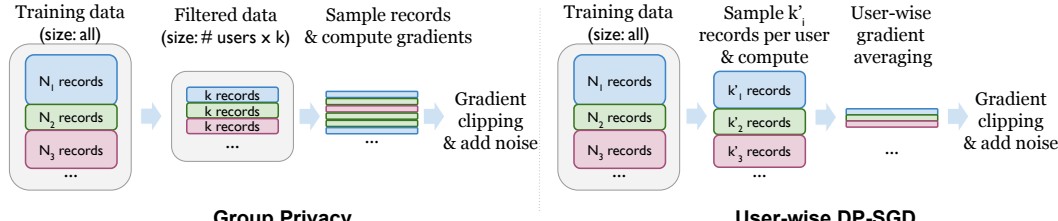

Figure 2: Illustration of how Group Privacy (left) and User-wise DP-SGD (right) preprocess training data, sample records, and calculate gradients at each training step.

## 2 User-level Differential Privacy Mechanisms

*Differential privacy (DP)* (Dwork et al., 2006a;b) is a mathematical framework for ensuring the privacy of individuals in datasets. Formally, a randomized algorithm $\mathcal{A}$ satisfies $(\varepsilon, \delta)$-*DP* if for any two neighboring datasets $\mathcal{D}$ and $\mathcal{D}'$, and any subset $\mathcal{S}$ of outputs, it holds for privacy parameters $\varepsilon \in \mathbb{R}_{>0}$ and $\delta \in [0, 1)$ that

$$\Pr[\mathcal{A}(\mathcal{D}) \in \mathcal{S}] \leq e^\varepsilon \cdot \Pr[\mathcal{A}(\mathcal{D}') \in \mathcal{S}] + \delta.$$

The definition of "neighboring datasets" is determined by the scope of privacy. For Example-level DP, two datasets are neighboring if one can be obtained from the other by adding/removing one text record. For User-level DP, we assume each user has a (variable) number of records, and two datasets are neighboring if one can be obtained from the other by adding/removing all the text belonging to a particular user.

### Algorithm 1: Group Privacy

1: **Input:** Initial model weights $\theta_0$, training set $D$ of $N$ users and $M$ records, learning rate $\eta$, iterations $T$, batch size $B$, privacy budget $\varepsilon$, number of records per user $k$, gradient norm bound $C$
/* **Stage 1. Compute noise multiplier** */
2: $\sigma \leftarrow$ GROUPPRIVACCOUNTING$(k, \varepsilon, \delta, M, B, T)$
/* **Stage 2. Prepare the dataset** */
3: $D' \leftarrow \varnothing$
4: **for** $i = 1, \dots, N$ **do**
5:     Sample $\{x_j, y_j\}_{j \in [k]}$, $k$ records for the $i$th user
6:     $D' \leftarrow D' \cup \{x_j, y_j\}_{j \in [k]}$
7: **end for**
/* **Stage 3. Training** */
8: **for** $t = 1, \dots, T$ **do**
9:     Randomly draw $\mathcal{B}_t$ from $D'$, a batch of $B$ records
10:     **for** $j = 1, \dots, B$ **do**
11:         $\mathbf{g}_{t,j} \leftarrow \nabla_{\theta_t} \ell(\theta_t, (x_j, y_j))$
/* **Clip gradient** */
12:         $\hat{\mathbf{g}}_{t,j} \leftarrow \mathbf{g}_{t,j} / \max\left(1, \frac{\|\mathbf{g}_{t,j}\|_2}{C}\right)$
13:     **end for**
/* **Aggregate and add noise** */
14:     $\tilde{\mathbf{g}}_t \leftarrow \frac{1}{B}\left(\sum_{j \in [B]} \hat{\mathbf{g}}_{t,j} + \mathcal{N}(0, \sigma^2 C^2 \mathbf{I}_d)\right)$
/* **Model update** */
15:     $\theta_{t+1} \leftarrow \theta_t - \eta \tilde{\mathbf{g}}_t$
16: **end for**

### Algorithm 2: User-wise DP-SGD

1: **Input:** Initial model weights $\theta_0$, training set $D$ of $N$ users and $M$ records, learning rate $\eta$, iterations $T$, user batch size $n$, privacy budget $\varepsilon$, number of records per user $\{k'_i\}_{i=1}^n$, gradient norm bound $C$
/* **Stage 1. Compute noise multiplier** */
2: $\sigma \leftarrow$ PRIVACCOUNTING$(\varepsilon, \delta, N, n, T)$
/* **Stage 2. Prepare the dataset** */
3: (User-wise DP-SGD directly uses $D$)
/* **Stage 3. Training** */
4: **for** $t = 1, \dots, T$ **do**
5:     Randomly draw $\mathcal{U}_t$, a batch of $n$ users
6:     **for** $i = 1, \dots, n$ **do**
7:         Sample $\{x_j, y_j\}_{j \in [k'_i]}$, $k'_i$ records for the $i$th user in $\mathcal{U}_t$
/* **Compute user-averaged gradient** */
8:         $\mathbf{g}_{t,i} \leftarrow \frac{1}{k'_i}\left(\sum_{j \in [k'_i]} \nabla_{\theta_t} \ell(\theta_t, (x_j, y_j))\right)$
/* **Clip gradient** */
9:         $\hat{\mathbf{g}}_{t,i} \leftarrow \mathbf{g}_{t,i} / \max\left(1, \frac{\|\mathbf{g}_{t,i}\|_2}{C}\right)$
10:     **end for**
/* **Aggregate and add noise** */
11:     $\tilde{\mathbf{g}}_t \leftarrow \frac{1}{n}\left(\sum_{j \in [n]} \hat{\mathbf{g}}_{t,i} + \mathcal{N}(0, \sigma^2 C^2 \mathbf{I}_d)\right)$
/* **Model update** */
12:     $\theta_{t+1} \leftarrow \theta_t - \eta \tilde{\mathbf{g}}_t$
13: **end for**

**Group Privacy.** Thanks to the composition property of the DP guarantee, an Example-level DP algorithm can be directly turned into a User-level DP algorithm for the case where each user contributes at most $k$ records. The simplest instantiation of such an approach says that if a mechanism $\mathcal{M}$ satisfies $(\varepsilon, \delta)$-DP then for all $k \geq 1$, the mechanism $\mathcal{M}'$, that limits contribution from each user to be at most $k$ (in an arbitrary manner), satisfies $(k\varepsilon, \delta\frac{e^{k\varepsilon}-1}{e^{\varepsilon}-1})$-user-level-DP (see, e.g., Vadhan (2017)). However, tighter group privacy bounds are known for the specific case of DP-SGD (Ganesh, 2024).[3] In Algorithm 1, we use GROUPPRIVACCOUNTING($k, \varepsilon, \delta, M, B, T$) to refer to any method that returns a $\sigma$ such that DP-SGD with $M$ records, batch size $B$, run for $T$ iterations and with each user contributing at most $k$ records, satisfies $(\varepsilon, \delta)$-DP. We provide more details on the specific accounting method we use in Appendix B.3.

**User-wise DP-SGD.** While Group Privacy can be applied to any underlying *record-level DP* algorithm, this flexibility comes at the potential cost of not utilizing the structure of the underlying algorithm directly. Here, we introduce a simple User-level DP algorithm, called User-wise DP-SGD, based on the classical DP-SGD algorithm (Abadi et al., 2016) for Example-level DP training. At each training step, User-wise DP-SGD samples a batch of users. For each user, it then samples several records, and computes the average gradients on those records[4]. The (average) gradients from the batch of users are then norm-clipped, mean-aggregated, and noised before passing to the optimizer. See Algorithm 2 for a formal algorithm description. We note the main modification of the original DP-SGD algorithm is user-wise sampling and clipping, which is similar to a combination of DP-SGD with FedAvg (McMahan et al., 2017) evaluated in some private federated learning tasks (McMahan et al., 2018; Kairouz et al., 2021). In this paper, we do not assume a federated learning setting, and allow the learner to inspect all user data to devise more sophisticated sampling strategies. PRIVACCOUNTING($\varepsilon, \delta, M, B, T$) here refers to any method that returns a $\sigma$ such that DP-SGD with $M$ records, batch size $B$, run for $T$ iterations, satisfies $(\varepsilon, \delta)$-DP. Since Algorithm 2 treats "users" as "records", it suffices to simply replace $M$ by $N$ (total number of users) and $B$ by $n$ (number of users sampled in a batch); see Appendix B.3 for more details. Comparing to Group Privacy, User-wise DP-SGD does not need to preprocess the dataset to restrict that each user contributes at most $k$ records. And we do not even need to assume each user's data consists of pre-defined records. For long form text, we can randomly sample contiguous chunks during training, creating more diverse records.

We also evaluate the algorithm in Asi & Liu (2024) (see Algorithm 3) that is based on User-wise DP-SGD with a more involved gradient estimation sub-procedure. In the sub-procedure, the algorithm detects if the average gradients for the users in the batch are well-concentrated, in other words, if they are close enough to each other in $\ell_2$-distance. If most of the gradients are well-concentrated, the outliers are removed, and the remaining gradients are mean-aggregated and noised. It is shown that less noise is needed in this case. In the other case, if there is no strong concentration, the algorithm halts and refuses to output anything meaningful. Under the assumption that all the records are i.i.d. from some underlying distribution, this algorithm can provably achieve the optimal privacy-utility trade-off.

## 3 Experimental Setup

**Datasets.** Our evaluation uses the Enron Email dataset (Klimt & Yang, 2004) and the BookSum dataset (Kryściński et al., 2022).

- The **Enron Email dataset** (Klimt & Yang, 2004) consists of approximately 500,000 emails generated by employees of the Enron Corporation. As these emails originate from real-world users and cover diverse topics, they may also contain privacy-sensitive information such as personally identifiable data, making this dataset a

---

[3]An earlier version of our work used the naive group privacy bound. However, following Charles et al. (2024), we have updated our approach to use the tighter accounting; for details see Appendix B.3.

[4]The number of records sampled for each user does not need to be the same for different users. However, for ease of implementation, we keep them the same in our experiments.

|  | **Enron Email** (Klimt & Yang, 2004) | **BookSum** (Kryściński et al., 2022) |
|---|---|---|
| # records for training | $240,173$ | $9,600$ |
| # records for evaluation | $39,086$ | $1,431$ |
| Avg. # records per privacy unit | 13.2 | 52.2 |
| Max # records per privacy unit | $5,049$ | 266 |
| Min # records per privacy unit | 1 | 1 |
| Median # records per privacy unit | 2.0 | 28.5 |
| Choice of $k$ | $\{2, 5, 10\}$ | $\{2, 5, 10, 20, 50\}$ |

Table 1: Sequence-level and privacy unit level dataset statistics for the Enron Email and BookSum datasets.

    suitable approximation of real-world user-level DP applications. We preprocess the dataset following the details provided in Appendix B.

- The **BookSum dataset** (Kryściński et al., 2022) contains human-written summaries of various books. Books and other literary works are typically protected by copyright laws, and training models on copyrighted material without permission from the copyright holders could raise legal issues (Henderson et al., 2023). While the summaries in the BookSum dataset may be considered transformative fair use of copyrighted books, applying DP training can further obfuscate the original data, potentially reducing the risk of memorizing and regurgitating copyrighted content from the source material.

**Privacy units.** For the Enron Email dataset, we consider each unique email sender as a single privacy unit. For BookSum, because the author name is not available in the dataset, we consider the book-level privacy instead. Example-level and privacy unit-level statistics for these two datasets are provided in Table 1.

**Models.** We use the pretrained GPT-2 small (125M) model (Radford et al., 2019) and LoRA (Hu et al., 2022) fine-tuning, following Yu et al. (2022).[5] Please see Appendix C.3 for results on larger GPT-2 models, and our analysis in Appendix C.1 showing that LoRA does not significantly impact model performance.

There is no established consensus on the optimal method for sampling records to train User-level DP mechanisms, whether using Group Privacy or User-wise DP-SGD. Therefore, in the following sections, we conduct a comprehensive evaluation exploring two key factors in this selection: (i) the record selection criteria, and (ii) the number of records to select for each privacy unit. Please see Appendix B for the detailed experimental setup and Appendix C.4 for the analysis of the computational overhead.

## 4 Group Privacy

In this section, we evaluate Group Privacy for User-level DP and explore design choices for selecting records for each user, including the data selection strategy (Section 4.1) and the number of records to be selected (Section 4.2).

### 4.1 Data selection strategy

We compare the following five methods for selecting records for each privacy unit:

- Random: Randomly select $k$ records from each privacy unit.

---

[5]We note that the choice of DP fine-tuning recipe is orthogonal to the user-level DP algorithms, as the latter can be seamlessly combined with any DP fine-tuning pipeline. We show in Appendix C.2 that we observe consistent results with a different fine-tuning method (Li et al., 2022).

|  | $\varepsilon = 1.0$ | $\varepsilon = 3.0$ | $\varepsilon = 8.0$ |
|---|---|---|---|
| Random | 35.23 | 31.45 | 28.51 |
| Longest | **34.95** | **31.32** | **28.23** |
| Shortest | 35.24 | 31.94 | 28.70 |
| Highest PPL | 35.28 | 31.53 | 28.40 |
| Lowest PPL | 35.28 | 31.62 | 28.58 |

(a) Enron Email

|  | $\varepsilon = 1.0$ | $\varepsilon = 3.0$ | $\varepsilon = 8.0$ |
|---|---|---|---|
| Random | **27.31** | 27.15 | **27.00** |
| Longest | 27.34 | 27.19 | 27.02 |
| Shortest | 27.40 | 27.18 | 27.03 |
| Highest PPL | 27.40 | 27.16 | 27.06 |
| Lowest PPL | 27.37 | **27.14** | 27.03 |

(b) BookSum

Table 2: Perplexity of Group Privacy with different sequence selection methods on the Enron Email dataset (a) and BookSum dataset (b) under varying privacy budget $\varepsilon$'s. Selecting the longest sequence consistently performs best for Enron Email. For BookSum, no single strategy dominates across all privacy budgets.

|  | $k = 2$ | $k = 5$ | $k = 10$ |
|---|---|---|---|
| $\varepsilon = 1.0$ | 35.67 | **34.26** | 35.23 |
| $\varepsilon = 3.0$ | 32.01 | **30.92** | 31.45 |
| $\varepsilon = 8.0$ | 30.13 | 28.74 | **28.51** |
| $\varepsilon = \infty$ | 23.00 | 19.88 | **19.53** |

(a) Enron Email

|  | $k = 2$ | $k = 5$ | $k = 10$ | $k = 20$ | $k = 50$ |
|---|---|---|---|---|---|
| $\varepsilon = 1.0$ | **27.30** | 27.34 | 27.31 | 27.54 | 27.46 |
| $\varepsilon = 3.0$ | **27.13** | 27.15 | 27.15 | 27.22 | 29.67 |
| $\varepsilon = 8.0$ | 27.08 | 27.05 | **27.00** | 27.07 | 27.22 |
| $\varepsilon = \infty$ | 26.07 | 25.75 | 25.62 | 25.53 | **25.43** |

(b) BookSum

Table 3: Perplexity of Group Privacy with different number of sequences per privacy unit (i.e., $k$) on the Enron Email dataset (a) and BookSum dataset (b) under varying privacy budget $\varepsilon$'s. Smaller values of $k$ generally achieve better results for smaller $\varepsilon$, while larger $\varepsilon$ favors larger $k$.

- Longest: Select $k$ records with the longest record length from each privacy unit.
- Shortest: Select $k$ records with the shortest record length from each privacy unit.
- Highest Perplexity (PPL): Select $k$ records with the highest perplexity, i.e., records that are most surprising to the (initial pre-trained) model.
- Lowest Perplexity (PPL): Select $k$ records with the lowest perplexity, i.e., records that are least surprising to the (initial pre-trained) model.

We fix $k = 10$ for both datasets and report the results in Table 2.

**Selection strategy matters for Enron Emails, but not for BookSum.** On the Enron Email dataset (Table 2a), selecting the longest records from each user yields the best performance across all privacy levels. This suggests that longer email records tend to be more informative and representative of each user's data, leading to better utility after private fine-tuning. For the BookSum dataset (Table 2b), the performance differences across selection methods are relatively small. Random selection performs best for $\varepsilon = 1.0$ and $8.0$, while selecting records with the lowest perplexity (least surprising to the model) works slightly better for $\varepsilon = 3.0$.

In general, the results indicate that simple heuristics like selecting the longest or shortest records can be effective strategies, sometimes outperforming more complex criteria like perplexity-based selection. The performance gap between different selection methods is more pronounced on the Enron Email dataset compared to BookSum. This could be due to the greater variation in record lengths and content in the email data versus book summary data (see Table 7 in Appendix B).

## 4.2 Number of selected records ($k$) per unit

The number of records ($k$) selected for each privacy unit is another important factor when applying group privacy. In group privacy, increasing $k$ makes both the noise and signal stronger, with opposing impact on the model's performance. On the one hand, a larger $k$ would result in higher noise due to the increased sensitivity of the privacy mechanism. On the other hand, using a larger $k$ also results in more diverse data being used for training,

|              | $\varepsilon = 1.0$ | $\varepsilon = 3.0$ | $\varepsilon = 8.0$ |
| --- | --- | --- | --- |
| Random chunk | **34.01** | **31.32** | **29.87** |
| Random       | 34.22 | 31.85 | 30.22 |
| Longest      | 34.47 | 31.76 | 30.54 |
| Shortest     | 35.68 | 32.91 | 31.19 |
| Highest PPL  | 34.36 | 31.65 | 30.16 |
| Lowest PPL   | 35.02 | 31.53 | 30.71 |

(a) Enron Email

|              | $\varepsilon = 1.0$ | $\varepsilon = 3.0$ | $\varepsilon = 8.0$ |
| --- | --- | --- | --- |
| Random chunk | **27.31** | **27.10** | **26.93** |
| Random       | 27.38 | 27.16 | 27.01 |
| Longest      | 27.38 | 27.17 | 27.02 |
| Shortest     | 27.42 | 27.16 | 27.02 |
| Highest PPL  | 27.41 | 27.18 | 27.04 |
| Lowest PPL   | 27.39 | 27.16 | 27.02 |

(b) BookSum

Table 4: Perplexity of User-wise DP-SGD with different sequence selection methods on the Enron Email dataset (a) and BookSum dataset (b) under varying privacy budgets (i.e., $\varepsilon$). Selecting the random chunks consistently performs best.

which can potentially improve the model's utility. Therefore, determining the optimal value of $k$ for utility is a trade-off between *the increased noise* and *the diversity of the training data*.

**Optimal $k$ balances noise from privacy and training data diversity.** To investigate the impact of $k$, we fix the data selection strategy to the random selection method and vary the value of $k$. We observe in Table 3 that for lower privacy budgets (i.e., $\varepsilon = 1.0$ or $3.0$), using a smaller $k$ is generally better, as the undesirable increase in noise when $k$ increases outweighs the benefits of diverse data. However, for larger $\varepsilon$, using a larger $k$ is usually preferred, as the high privacy budget can potentially tolerate slightly more noise being introduced. In such cases, the diversity of the training data becomes more beneficial for larger $k$, as evidenced by lower perplexity for larger $k$'s when $\varepsilon = \infty$ (non-private setting)[6]. We also observe similar results on GPT-2 medium and large models, as detailed in Appendix C.3.

## 5   User-wise DP-SGD

In this section, we evaluate User-wise DP-SGD and explore design choices for selecting records for each user. We discuss the data selection strategy (Section 5.1) and the number of records to be selected (Section 5.2). We also analyze the applicability of the advanced User-wise DP-SGD method proposed by Asi & Liu (2024) based on a more advanced gradient estimation sub-procedure.

### 5.1   Data selection strategy

Unlike Group Privacy that requires preprocessing to limit each user's contribution to $k$ records, User-wise DP-SGD imposes no such constraint. This allows exploring data selection strategies beyond record boundaries. Therefore, in addition to the five selection strategies discussed in Section 4.1, we also explore a method called **Random Chunk**. Specifically, for each user, we concatenate all their records into a single document, and randomly fetch a chunk of the maximum record length (as defined by the model) from it.

As shown in Table 4, **Random Chunk** consistently outperforms other methods across datasets and privacy budgets $\epsilon$. This benefit likely stems from increased data diversity during training and the ability to span record boundaries.

### 5.2   Number of selected records ($k$) per unit

**User-wise DP-SGD favors larger $k$, with diminishing returns.**   In contrast to Group Privacy (Section 4.2), increasing $k$ monotonically improves User-wise DP-SGD's utility by providing more diverse data per user. However, larger $k$ also incurs slightly higher computation (Appendix C.5). The benefit diminishes as $k$ grows, so for practical deployment, choosing a

---

[6]Note that for both methods, we report results for $\varepsilon = \infty$ following their corresponding algorithms in Algorithm 1 and Algorithm 2 while bypassing the stage of aggregation and noise addition.

|  | $k = 2$ | $k = 5$ | $k = 10$ |
|---|---|---|---|
| $\varepsilon = 1.0$ | 34.37 | 35.02 | **34.01** |
| $\varepsilon = 3.0$ | 31.31 | 31.92 | **31.32** |
| $\varepsilon = 8.0$ | 30.12 | 30.29 | **29.87** |
| $\varepsilon = \infty$ | 22.87 | 19.67 | **18.64** |

|  | $k = 2$ | $k = 5$ | $k = 10$ | $k = 20$ | $k = 50$ |
|---|---|---|---|---|---|
| $\varepsilon = 1.0$ | 27.62 | 27.38 | 27.38 | 27.35 | **27.33** |
| $\varepsilon = 3.0$ | 27.29 | 27.19 | 27.18 | **27.17** | 27.17 |
| $\varepsilon = 8.0$ | 27.11 | 27.02 | 27.01 | **27.00** | 27.01 |
| $\varepsilon = \infty$ | 24.38 | 24.22 | 24.12 | 24.11 | **24.09** |

(a) Enron Email          (b) BookSum

Table 5: Perplexity of User-wise DP-SGD with different number of sequences per privacy unit (i.e., $k$) on the Enron Email dataset (a) and BookSum dataset (b) under varying privacy budget $\varepsilon$'s. Using a larger value of $k$ consistently improves performance.

sufficiently large $k$ will balance utility and efficiency. For instance, on the BookSum dataset, $k = 10$ appears to achieve the best trade-off between these two factors.

### 5.3 Analysis of the advanced User-wise DP-SGD

As discussed in Section 2, Asi & Liu (2024) proposed a User-wise DP-SGD algorithm that achieves the state-of-the-art excess rate and requires injecting less noise if per-user averaged gradients in a batch exhibit low variance. Specifically, recall that User-wise DP-SGD has a clipping norm parameter $C$, and the noise added is proportional to $C$. Asi & Liu (2024) showed that when most of the gradients are within $\ell_2$-distance $\tau$ of each other, adding noise proportional to $\tau$ suffices for privacy purposes. Under the assumption that the records are drawn i.i.d. from some underlying distribution, Asi & Liu (2024) proved that this mechanism can achieve the optimal rate.

However, it remains unclear how applicable this algorithm, and other such theoretically motivated ones, are to real-world problems where the i.i.d. assumption might not be true, and the number of records per user is relatively small (e.g., $k \leq 50$). In order to answer this question, we measure the noise levels of standard User-wise DP-SGD and Asi & Liu (2024)'s method under different values of $\tau$, controlling the required gradient concentration. Figure 3 shows that for most $k$ and $\epsilon$, Asi & Liu (2024)'s noise is lower than User-wise DP-SGD's only if $\tau/C < 0.1$, making it highly unlikely to enter the advanced noising step (lines 10–15 in Algorithm 3). Despite its potential benefits, this method relies on high user gradient concentration, which does not hold in our settings.

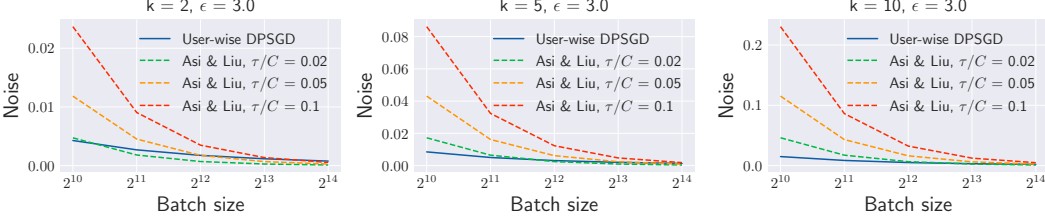

Figure 3: Effective noise of User-wise DP-SGD and the advanced method proposed by Asi & Liu (2024) under different numbers of records per user ($k$), with $\varepsilon = 3.0$. As shown, for Asi & Liu (2024) to yield lower noise than standard User-wise DP-SGD, the ratio between the concentration factor $\tau$ and the clipping norm $C$ must be smaller than 0.1. However, in the evaluated datasets, $\tau/C$ is around 1. Figure 7 presents results for different $\varepsilon$'s.

## 6 Comparing Group Privacy and User-wise DP-SGD

Finally, we compare the performance of Group Privacy and User-wise DP-SGD under the same privacy budget. For Group Privacy, we use **Longest** as its data selection strategy, and for each $\varepsilon$, we use its corresponding best $k$, as reported in Table 3. For User-wise DP-SGD, we use **Random Chunk** for data selection, and use $k = 10$, as it achieves the best trade-off between utility and efficiency. We vary the other hyperparameters, including learning rate, batch

|  | Group Privacy | User-wise DP-SGD |
|---|---|---|
| $\varepsilon = 0.5$ | 35.28 | **35.08** |
| $\varepsilon = 1.0$ | 33.50 | **33.36** |
| $\varepsilon = 3.0$ | 31.01 | **30.98** |
| $\varepsilon = 8.0$ | 28.51 | **28.32** |

(a) Enron Email

|  | Group Privacy | User-wise DP-SGD |
|---|---|---|
| $\varepsilon = 0.5$ | 27.50 | **27.43** |
| $\varepsilon = 1.0$ | 27.41 | **27.29** |
| $\varepsilon = 3.0$ | 27.17 | **27.15** |
| $\varepsilon = 8.0$ | **27.00** | 27.02 |

(b) BookSum

Table 6: Perplexity of Group Privacy and User-wise DP-SGD under varying privacy budgets ($\varepsilon$). User-wise DP-SGD generally outperforms Group Privacy, especially for smaller privacy budgets.

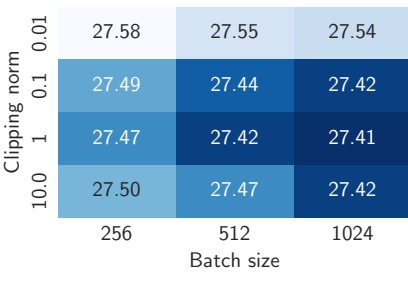

(a) Group Privacy

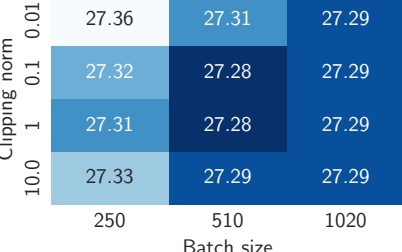

(b) User-wise DP-SGD

Figure 4: Perplexity of Group Privacy (a) and User-wise DP-SGD (b) on the BookSum dataset with a privacy budget of $\varepsilon = 1.0$, under varying clipping norms and batch sizes. Using larger batch sizes generally improves the performance for both methods. However, Group Privacy exhibits higher sensitivity to clipping norm variations compared to User-wise DP-SGD.

size, number of training iterations, and the clipping norm, and report the best results for each mechanism in Table 6.

**User-wise DP-SGD outperforms Group Privacy.** At the mechanism level, User-wise DP-SGD consistently outperforms Group Privacy, especially for smaller privacy budgets. This is potentially due to User-wise DP-SGD allowing for more diverse data being selected in different minibatches during training, while Group Privacy already constrains the training data to be a subset of user records during the preprocessing stage. Additionally, for some users who have fewer than $k$ records, our implementation of Group Privacy will perform resampling to obtain $k$ records. This repetition may also negatively impact the model's performance. We acknowledge this as a potential limitation of our evaluation that could be addressed in future work. However, it is worth noting that Group Privacy requires minimal changes to implement on top of existing record-level private training, whereas User-wise DP-SGD may require modifications to the data loading pipeline. In this regard, practitioners may need to consider the implementation difficulty when choosing between these two methods.

**Sensitivity to clipping norms.** We specifically investigate the sensitivity of both mechanisms to clipping norms, as reported in Figure 4. As shown, Group Privacy is generally more sensitive to clipping norms compared to User-wise DP-SGD. This is because the clipping operation is performed on the gradient of each individual record for the Group Privacy method (see Algorithm 1). In contrast, for User-wise DP-SGD, the clipping is applied to the averaged gradients, which ideally exhibit less variance, making it less sensitive to clipping norms (see Algorithm 2).

## 7 Related Work

**Record-level Differential Privacy.** Differential privacy (DP) (Dwork et al., 2006b;a) is a widely adopted framework that formally bounds the privacy leakage of individual user information while still enabling the analysis of population-level patterns. When training deep neural networks with DP guarantees, the go-to algorithm is Differentially Private

Stochastic Gradient Descent (DP-SGD), introduced in the seminal work of Abadi et al. (2016). This algorithm operates by clipping the contributions of gradients on a per-record basis and injecting noise into the average gradient updates during each iteration of SGD.

Previous research has applied DP-SGD for language models (Ponomareva et al., 2023), primarily focusing on the record level. These applications include work in natural language understanding (Li et al., 2022; Yu et al., 2021; 2022), natural language generation (Anil et al., 2022), in-context learning (Tang et al., 2024; Wu et al., 2023; Duan et al., 2023), prompt tuning (Li et al., 2023b), synthetic text generation (Al Aziz et al., 2021; Yue et al., 2023), and retrieval-augmented language models (Huang et al., 2023).

**User-level Differential Privacy.** McMahan et al. (2018); Kairouz et al. (2021) studied user-level DP training of LSTM based language models in the federated learning setting. In contrast, we consider a general central training setting for more flexible samplers. Moreover, we studied modern transformer-based language models. More broadly, several fundamental and important problems have been actively studied under user-level DP in recent years, e.g., machine learning (Wang et al., 2019), learning discrete distributions (Liu et al., 2020), histogram estimation (Amin et al., 2019), Structured Query Language (SQL) (Wilson et al., 2020), number of users required with multiple items per user (Ghazi et al., 2021), stochastic convex optimization (Levy et al., 2021; Bassily & Sun, 2023; Ghazi et al., 2023a; Asi & Liu, 2024) and reduction from item-level to user-level DP (Ghazi et al., 2023b). As mentioned, a common assumption in the theoretical results above is that the data is drawn i.i.d. from an underlying distribution. This assumption is too restrictive, and it would be interesting to propose more realistic assumptions under which we can still gain some theoretical insights.

We note that an independent and concurrent work of Charles et al. (2024) also examines the empirical performance of user-level DP methods for fine-tuning LLMs. The key findings that User-wise DP-SGD usually outperforms Group Privacy in terms of utility are generally consistent between the two studies. Notably, their work provides valuable insights by offering an explicit methodology for selecting the group size in both mechanisms, whereas ours mainly performs ablations of various sizes. Moreover, our work differs from theirs in that we explore a broader design space for data selection strategies under various User-level DP mechanisms. In this expanded exploration, we observe substantial performance gaps between suboptimal and better data selection methods. In addition, their evaluation focuses on the larger StackOverflow and news datasets using a single 350M-parameter model while our work utilizes books and email datasets and considers models of various sizes. We also provide case studies for the User-level DP algorithm with the state-of-the-art excess rate.

## 8    Conclusions

In this work, we study language model fine-tuning with DP guarantees. Since each user generally contributes multiple text records to the training corpus, to ensure equal privacy guarantees across users, the privacy unit needs to be set at the user level. We identify and systematically evaluate two mechanisms, Group Privacy and User-wise DP-SGD, for achieving user-level DP when fine-tuning large language models on natural language generation tasks. For each mechanism, we investigate data selection strategies to improve the trade-off between privacy and utility.

While our work provides valuable insights, we acknowledge a few limitations and suggest directions for future research. First, in our implementation of User-wise DP-SGD, we fixed the value of $k$ (the number of gradients to sample per user) for simplicity. Exploring more dynamic choices of $k$ for different users may yield better utility while preserving privacy guarantees. Second, since User-wise DP-SGD can perform data selection on the fly, it will be interesting to investigate how other data selection methods, such as those based on gradient information or active learning, might impact the utility. Finally, while we focus on language generation, user-level DP has broader applications motivating analysis in domains like recommendation systems and structured prediction.

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

# Appendices

## A  User-Level Algorithm of Asi & Liu (2024)

We describe the algorithm in Asi & Liu (2024) using our notation for completeness. They assume each user owns the same amount of record and use the full batch size (use all dataset in each step). We modify the algorithm by drawing $k'_i$ records for the $i$th user in the following Pseudocode.

---

**Algorithm 3** User-wise DP-SGD with filter

---

1: **Input:** Initial model weights $\theta_0$, training set $D$ of $N$ users and $M$ records, learning rate $\eta$, iterations $T$, privacy budget $\varepsilon$, number of records per user $\{k'_i\}_{i=1}^n$, concentration threshold $\tau$
   /* **Stage 1. Compute noise multiplier** */
2: $\sigma \leftarrow \text{PRIVACCOUNTING}(\varepsilon/2, \delta/2, N, N, T)$
   /* **Stage 2. Training** */
3: **for** $t = 1, \ldots, T$ **do**
4:   **for** $i = 1, \ldots, N$ **do**
5:     Sample $\{x_j, y_j\}_{j \in [k'_i]}$, $k'_i$ records for the $i$th user in $D$
       /* Compute user-averaged gradient */
6:     $\mathbf{g}_{t,i} \leftarrow \frac{1}{k'_i} \left( \sum_{j \in [k'_i]} \nabla_{\theta_t} \ell(\theta_t, (x_j, y_j)) \right)$
7:   **end for**
     /* Compute the concentration scores */
8:   Compute

$$s_t^{\text{conc}}(D, \tau) := \frac{1}{N} \sum_{i,i'} \mathbb{1}(\|\mathbf{g}_{t,i} - \mathbf{g}_{t,i'}\| \leq \tau). \tag{1}$$

   /* Run the AboveThreshold with the concentration scores.  If it passes, remove the outliers */
9:   **if** AboveThreshold$(s_t^{\text{conc}}, \frac{\varepsilon}{2}, 4N/5) = \top$ **then**
10:     Set $\mathcal{B}_t = \varnothing$
11:     Set $f_{t,i} = \sum_{i'} \mathbb{1}(\|\mathbf{g}_{t,i} - \mathbf{g}_{t,i'}\| \leq 2\tau)$

12:     Add $i^{\text{th}}$ user to $\mathcal{B}_t$ with probability $p_{t,i}$ for $p_{t,i} = \begin{cases} 0 & f_{t,i} < N/2 \\ 1 & f_{t,i} \geq 2N/3 \\ \frac{f_{t,i} - N/2}{N/6} & o.w. \end{cases}$

       /* Aggregate and add noise for the remaining users */
13:     Let $\hat{\mathbf{g}}_t = \frac{1}{|\mathcal{B}_t|} \sum_{i \in \mathcal{B}_t} \mathbf{g}_{t,i}$ if $\mathcal{B}_t$ is not empty, and 0 otherwise
14:     $\tilde{\mathbf{g}}_t \leftarrow \hat{\mathbf{g}}_t + \nu_t$, where $\nu_t \sim \mathcal{N}(0, 8\tau^2 \log(e^\varepsilon T/\delta)\sigma^2/N^2 \mathbf{I}_d)$
       /* Model update */
15:     $\theta_{t+1} \leftarrow \theta_t - \eta \tilde{\mathbf{g}}_t$
16:   **else**
17:     **Halt**. /* Halt the algorithm if it does not pass. */
18:   **end if**
19: **end for**

---

AboveThreshold is a classic algorithm used in DP literature. See Algorithm 1 in Asi & Liu (2024) for more details and references.

## B Experimental details

### B.1 Dataset details

**Data preprocessing for the Enron Email dataset.** To prepare the Enron Email dataset (Klimt & Yang, 2004) for our evaluation of user-level DP implementations, we first parse each record, into DATE, FROM (sender), TO (receiver), SUBJECT, and BODY, using regular expressions. Note that for each of the "CC" or "BCC" addresses on the email, we treat it as a new receiver. Specifically, we will create a new record with the same (DATE, FROM, SUBJECT, BODY), but a different TO (i.e., receiver). We then perform in-email deduplication to attain a dataset with higher quality: We only keep records whose maximum repetition of any 20-gram is 1. Forwarded emails contain the email body of the forwarded messages; this complicates the privacy accounting process. Therefore, for simplicity, we remove forwarded emails. After preprocessing, the final dataset contains 240, 173 emails.

**Distribution of number of records per privacy unit.** Figure 5 presents the distribution of number of records per privacy unit in both datasets.

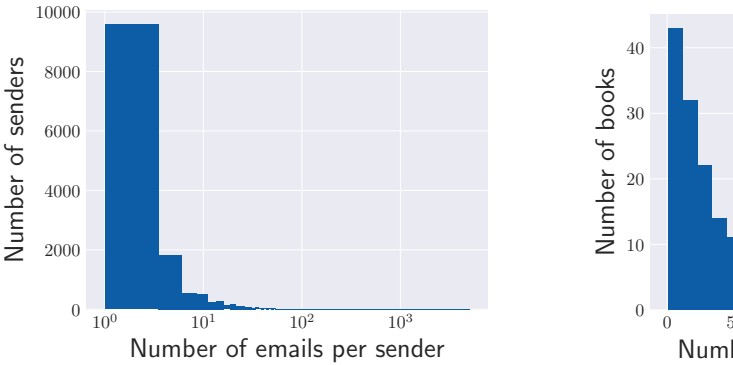
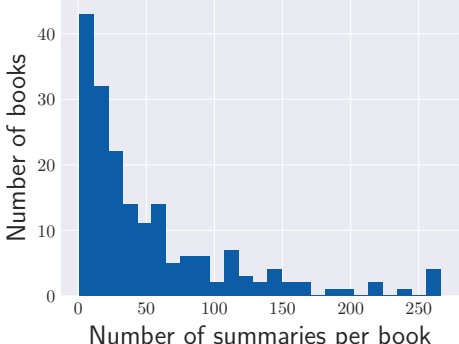

Figure 5: Distribution of number of records per privacy unit in Enron Email (left) and BookSum (right).

**Statistics after data selection.** For reference, Table 7 reports some statistics of the selected records after applying different data selection methods discussed in Section 4.1.

### B.2 Model details

Our experiments primarily use the pretrained GPT-2 small model (Radford et al., 2019), which has 125 million trainable parameters. To assess the generalizability of our findings, we also present results with GPT-2 models of different sizes in Appendix C.3. For each evaluated dataset, we fine-tune the model for the language modeling task, using LoRA (Hu et al., 2022) with a rank of 32. As shown in Appendix C.1, LoRA does not significantly impact model performance. Model performance is measured using perplexity, a standard evaluation metric for language models.

### B.3 Privacy Accounting

We perform privacy accounting assuming that the batches are constructed using Poisson subsampling, where each batch is independently sampled by including each record / user independently with a certain probability, even though our implementation involves constructing batches by going through users/records in a random order. We note that this assumption has been a common practice in the privacy accounting of DP-SGD, starting with the work of Abadi et al. (2016). Recent work by Chua et al. (2024) points out that there can be substantial gaps between the privacy guarantees of DP-SGD when using Poisson subsampling vs. shuffling; however, since we make the Poisson subsampling assumption

|  | Avg. length | Avg. PPL | Fraction of selected sequences w/ length < max_seq_len |
|---|---|---|---|
| Random | 240.67 | 214.67 | 0.73 |
| Longest | 274.91 | 201.14 | 0.69 |
| Shortest | 222.28 | 278.75 | 0.76 |
| Highest PPL | 233.27 | 1123.77 | 0.74 |
| Lowest PPL | 247.18 | 188.92 | 0.72 |

(a) Enron Email

|  | Avg. length | Avg. PPL | Fraction of selected sequences w/ length < max_seq_len |
|---|---|---|---|
| Random | 557.49 | 34.37 | 0.61 |
| Longest | 817.93 | 33.79 | 0.30 |
| Shortest | 338.43 | 41.99 | 0.80 |
| Highest PPL | 483.79 | 50.32 | 0.66 |
| Lowest PPL | 539.52 | 25.94 | 0.65 |

(b) BookSum

Table 7: Comparison of the average sequence length, perplexity, and fraction of selected sequences with length smaller than the maximum allowed sequence length, after applying different data selection techniques to the (a) Enron Email dataset and (b) BookSum dataset.

for both Group Privacy and User-wise DP-SGD, we consider our evaluation to still be informative in a relative sense even in the presence of this gap.

The privacy analysis of DP-SGD using Poisson subsampling with $M$ records and expected batch size $B$, run for $T$ steps is performed by analyzing the $T$-fold composition of the Poisson subsampled Gaussian mechanism with subsampling probability of $B/M$. It is possible to numerically compute the smallest $\sigma$, up to some tolerance, such that this mechanism satisfies $(\varepsilon, \delta)$-DP using algorithms based on privacy loss distributions (PLD) (Koskela et al., 2021; Gopi et al., 2021; Doroshenko et al., 2022), using various open source libraries Microsoft. (2021); Prediger & Koskela (2020). In particular, we use the implementation in the dp_accounting library (Google's DP Library., 2020). We use PRIVACCOUNTING$(\varepsilon, \delta, M, B, T)$ to refer to this smallest $\sigma$.

**PRIVACCOUNTING for Algorithm 2** Under the assumption of Poisson subsampling of users, the same privacy analysis is applicable with "number of records" $M$ replaced by "number of users" $N$ and "batch size" $B$ replaced by "user batch size" $n$, and thus, the optimal noise parameter is given as PRIVACCOUNTING$(\varepsilon, \delta, N, n, T)$.

**GROUPPRIVACCOUNTING for Algorithm 1.** Ganesh (2024) showed that the privacy analysis of a single step of DP-SGD under addition/removal of $k$ records under Poisson subsampling is given by considering the Gaussian mechanism with sensitivity that is randomly drawn from the Binomial distribution $\text{Bin}(k, B/M)$ referred to as the "Mixture of Gaussians (MoG)" mechanism. The privacy analysis of the entire DP-SGD can thus be performed by analyzing the $T$-fold composition of this MoG mechanism, which can be done in a tight manner up to numerical accuracy using privacy loss distributions. We use the implementation available in the dp_accounting library.

# C More results

## C.1 LoRA v.s. Full fine-tuning

Table 8 presents the perplexity scores for full fine-tuning and Low-Rank Adaptation (LoRA) (Hu et al., 2022) fine-tuning on the Enron Email and BookSum datasets. As shown, applying LoRA fine-tuning has a relatively minor impact on the model's utility.

|  | Enron Email | BookSum |
|---|---|---|
| Full fine-tuning | 18.53 | 23.79 |
| LoRA, $r = 64$ | 18.83 | 24.07 |
| LoRA, $r = 32$ | 19.05 | 24.18 |
| LoRA, $r = 16$ | 19.39 | 24.31 |

Table 8: Comparison of perplexity scores for full fine-tuning and Low-Rank Adaptation (LoRA) fine-tuning with different rank values on the Enron Email and BookSum datasets.

## C.2 Compatibility with a different DP fine-tuning method

Note that the choice of DP fine-tuning recipe is orthogonal to the user-level DP algorithms, as the latter can be seamlessly combined with any DP fine-tuning pipelines.

We also rerun our evaluation using the pipeline from Li et al. (2022) on the Enron Email dataset. Consistent with our previous findings, random chunking works best for User-wise DP-SGD, and longest sequence selection works best for Group Privacy. The performance of both mechanisms, using their respective optimal selection strategies, is shown in Table 9 and aligns with our findings in the main paper.

|  | Group Privacy | User-wise DP-SGD |
|---|---|---|
| $\epsilon = 1.0$ | 33.44 | **33.21** |
| $\epsilon = 3.0$ | 31.08 | **30.76** |
| $\epsilon = 8.0$ | 28.34 | **28.07** |

Table 9: Perplexity of Group Privacy and User-wise DP-SGD under different privacy budgets ($\varepsilon$) on the Enron Email dataset, using the DP-fine-tuning method from Li et al. (2022). User-wise DP-SGD generally outperforms Group Privacy.

## C.3 Results under different models

We further report the performance of varying the number of records per privacy unit ($k$) for Group Privacy and User-wise DP-SGD under different sizes of GPT-2 models in Table 10 and Table 11, respectively. The observed trends are consistent across different model sizes.

|  | $k = 2$ | $k = 5$ | $k = 10$ | $k = 20$ | $k = 50$ |
|---|---|---|---|---|---|
| $\varepsilon = 1.0$ | **27.31** | 27.34 | 27.45 | 27.56 | 27.49 |
| $\varepsilon = 3.0$ | **27.14** | 27.15 | 27.17 | 27.24 | 29.68 |
| $\varepsilon = 8.0$ | 27.08 | 27.05 | **27.02** | 27.07 | 27.22 |

(a) GPT2-small

|  | $k = 2$ | $k = 5$ | $k = 10$ | $k = 20$ | $k = 50$ |
|---|---|---|---|---|---|
| $\varepsilon = 1.0$ | **21.83** | 21.91 | 21.99 | 22.09 | 22.87 |
| $\varepsilon = 3.0$ | 21.70 | 21.73 | **21.68** | 21.72 | 22.05 |
| $\varepsilon = 8.0$ | 21.61 | 21.60 | **21.52** | 21.53 | 21.68 |

(b) GPT2-medium

|  | $k = 2$ | $k = 5$ | $k = 10$ | $k = 20$ | $k = 50$ |
|---|---|---|---|---|---|
| $\varepsilon = 1.0$ | **19.28** | **19.28** | 19.32 | 19.39 | 20.04 |
| $\varepsilon = 3.0$ | 19.21 | **19.18** | **19.18** | 19.22 | 19.74 |
| $\varepsilon = 8.0$ | 19.12 | 19.03 | **18.99** | 19.00 | 19.31 |

(c) GPT2-large

Table 10: Perplexity of Group Privacy with different number of sequences per privacy unit (i.e., $k$) on the Enron Email dataset (a) and BookSum dataset (b), under varying privacy budget $\varepsilon$'s, using different sizes of GPT-2 models. Smaller values of $k$ generally achieve better results for smaller $\varepsilon$, while larger $\varepsilon$ favors larger $k$.

|  | $k = 2$ | $k = 5$ | $k = 10$ | $k = 20$ | $k = 50$ |
|---|---|---|---|---|---|
| $\varepsilon = 1.0$ | 27.62 | 27.38 | 27.38 | 27.35 | **27.33** |
| $\varepsilon = 3.0$ | 27.29 | 27.19 | 27.16 | 27.17 | **27.17** |
| $\varepsilon = 8.0$ | 27.11 | 27.02 | 27.01 | 27.00 | **27.01** |

(a) GPT2-small

|  | $k = 2$ | $k = 5$ | $k = 10$ | $k = 20$ | $k = 50$ |
|---|---|---|---|---|---|
| $\varepsilon = 1.0$ | 21.88 | 21.80 | 21.77 | **21.75** | **21.75** |
| $\varepsilon = 3.0$ | 21.61 | 21.52 | 21.54 | 21.52 | **21.51** |
| $\varepsilon = 8.0$ | 21.50 | 21.47 | 21.44 | **21.43** | **21.43** |

(b) GPT2-medium

|  | $k = 2$ | $k = 5$ | $k = 10$ | $k = 20$ | $k = 50$ |
|---|---|---|---|---|---|
| $\varepsilon = 1.0$ | 19.22 | 19.21 | **19.19** | **19.19** | **19.19** |
| $\varepsilon = 3.0$ | 19.13 | 19.08 | 19.06 | **19.05** | **19.05** |
| $\varepsilon = 8.0$ | 18.99 | 18.93 | 18.93 | **18.92** | **18.92** |

(c) GPT2-large

Table 11: Perplexity of User-wise DP-SGD with different number of sequences per privacy unit (i.e., $k$) on the Enron Email dataset (a) and BookSum dataset (b), under varying privacy budget $\varepsilon$'s, using different sizes of GPT-2 models. Using a larger value of $k$ consistently improves performance.

### C.4 Computational overhead

The main overhead of Group Privacy training methods is introduced by the per-example gradient clipping and noise addition (lines 11–13 in Algorithm 1). For the User-wise DP-SGD method, the primary overhead stems from the user-wise data sampling (line 5 in Algorithm 2), as well as per-user gradient clipping and noise addition (lines 8–9 in Algorithm Algorithm 2).

We record the training time (in seconds) per step for different models and training methods with $k = 10$ on the Enron Email dataset in Table 12. The results are obtained using LoRA fine-tuning with $r = 32$ and a batch size of 1024, averaged across 100 training steps, on a single NVIDIA A100-80G GPU. As shown, both methods introduce very little computational overhead (4%–9%) compared to non-private training, regardless of the model size.

| Model | Non-private training | Group privacy | User-wise DP-SGD |
|---|---|---|---|
| GPT2-small | 11.95 | 12.70 (1.06×) | 12.82 (1.07×) |
| GPT2-medium | 30.90 | 32.00 (1.04×) | 32.85 (1.06×) |
| GPT2-large | 62.57 | 64.80 (1.04×) | 68.45 (1.09×) |

Table 12: Comparison of per-step runtime (seconds) for non-private training, and training with Group Privacy and User-wise DP-SGD on GPT-2 models of varying sizes.

### C.5 Cost of $k$ in User-wise DP-SGD

Increasing $k$ in Group Privacy automatically results in larger datasets, and thus higher training time. We also show in Figure 6 that increasing $k$ in User-wise DP-SGD also incurs longer running time, due to the cost of sampling more records per training step.

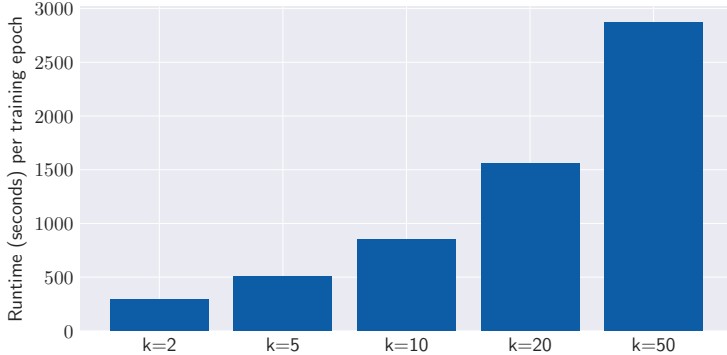

Figure 6: Runtime per training epoch under different $k$'s for BookSum dataset with User-wise DP-SGD.

### C.6 More results for comparing User-wise DP-SGD and Asi & Liu (2024)

Figure 7 compares the noise of User-wise DP-SGD and the advanced method proposed by Asi & Liu (2024) under different $\varepsilon$'s. Still, for Asi & Liu (2024) to yield lower noise than standard User-wise DP-SGD, the ratio between the concentration factor $\tau$ and the clipping norm $C$ must be smaller than 0.1. However, in the evaluated datasets, $\tau/C$ is around 1.

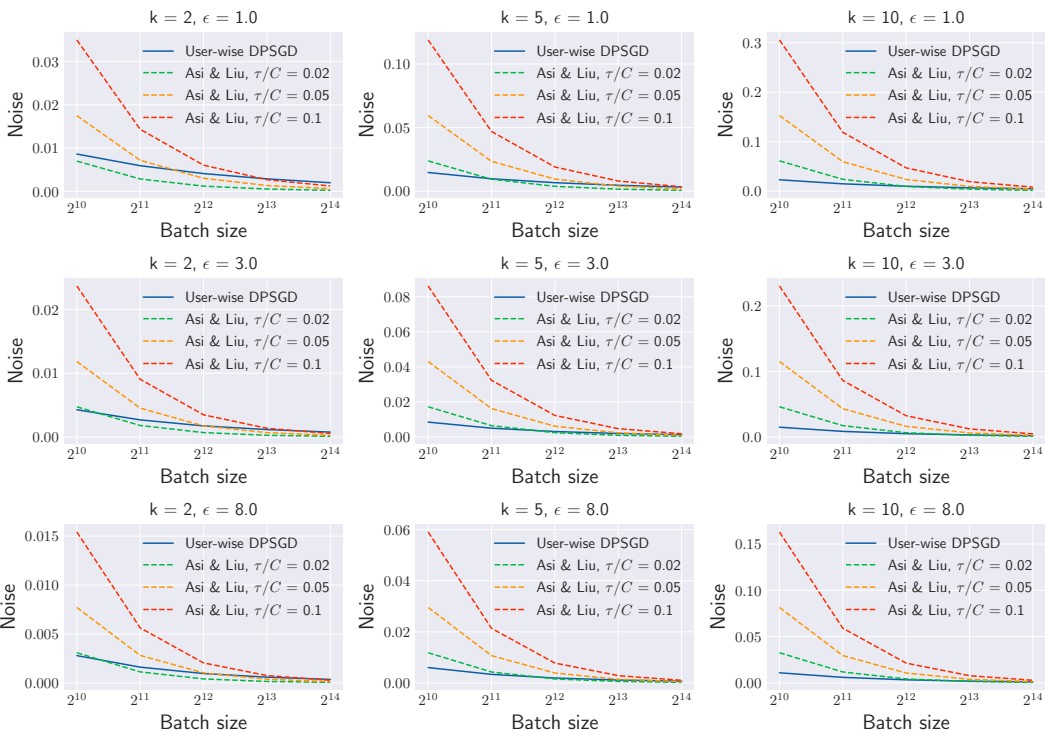

Figure 7: Effective noise of User-wise DP-SGD and the advanced method proposed by Asi & Liu (2024) under different $\epsilon$'s and numbers of records per user ($k$). As shown, for Asi & Liu (2024) to yield lower noise than standard User-wise DP-SGD, the ratio between the concentration factor $\tau$ and the clipping norm $C$ must be smaller than 0.1. However, in the evaluated datasets, $\tau/C$ is around 1.

