# OpenReview forum: "Mind the Privacy Unit! User-Level Differential Privacy for Language Model Fine-Tuning"
_colmweb.org/COLM/2024/Conference — COLM_

### Official Review · Reviewer_sVEv · 2024-05-10

**Rating:** 6
**Confidence:** 4
**Ethics Flag:** 1

**Summary:**

The paper delves into the issue of privacy concerns when fine-tuning large language models (LLMs) on sensitive data. The paper introduces the concept of User-level Differential Privacy (DP) as a solution to ensure uniform privacy protection across users, especially in scenarios where contributions per user vary. By focusing on two mechanisms, Group Privacy and User-wise DP-SGD, the study evaluates the design choices such as data selection strategies and parameter tuning to strike a balance between privacy and utility.
Overall, the paper provides a new perspect of user-wise DP for LLM fine-tuning on natural language generation tasks, shedding light on the importance of considering user privacy guarantees in addition to example-level privacy. However, the solution is somehow lacking of novelty. Moreover, the comparison of proposed approach is not thorough.

**Questions To Authors:**

1). More thorugh comparisons to other SOTAs of DP.

2). What if the user info is not available for different samples?

**Reasons To Accept:**

1). The work is easy to follow and well-organized: The paper on User-Level Differential Privacy for Language Model Fine-Tuning excels in its clarity and organization, making it easy for readers to follow the research methodology, findings, and implications. The paper is structured in a logical manner, with clear section headings and transitions that guide the reader through the exploration of User-level DP mechanisms. By presenting a systematic evaluation of privacy protection strategies in a coherent and accessible manner, the paper ensures that both experts and non-experts in the field can grasp the significance of the research .

2). The problem of privacy protection is an important issue: The paper underscores the critical importance of privacy protection in the context of fine-tuning large language models on sensitive data. With the proliferation of advanced machine learning models and the potential risks of memorization when handling personal or confidential information, ensuring robust privacy guarantees has become a paramount concern in the field of artificial intelligence and data science.

**Reasons To Reject:**

1). The solution is somehow straightforward building upon the existing groupwise DPSGD which limits the contribution of the submission.

2).  the comparison of proposed approach is not thorough. Many other DP approaches are not compared such as: LARGE LANGUAGE MODELS CAN BE STRONG DIFFERENTIALLY PRIVATE LEARNERS
https://openreview.net/pdf?id=bVuP3ltATMz
To name a few.

3). Typos: Page 3: a Example-level==>an Example-level

4). The method needs the present/availablility of user info, which might be missing in many real applications.

---

> ### Author Rebuttal · Authors · 2024-05-30
>
> ### Q1 Limited contribution: straightforward solution building upon existing method
>
> **A**: While our work builds on existing methods, our main contribution is to provide a comprehensive understanding of the pros and cons of group privacy and user-wise DP-SGD through systematic evaluation. We tested two datasets, three models, various DP budgets, different examples per user, and multiple data selection methods.
>
> Our proposed data selection strategies, such as random chunk selection that crosses sequence boundaries, are **well-motivated**. They aim to maximize the preservation of per-user information to achieve a better privacy-utility tradeoff and have proven to be **effective**, improving the Enron Email perplexity from 35.68 to 34.01 w/ ε=1.0.
>
> ### Q2 Comparison w/ other DP approaches
> > e.g. Large Language Models Can Be Strong Differentially Private Learners. ICLR’22
>
> **A**: Please note that the choice of DP fine-tuning recipe is **orthogonal** to the user-level DP algorithms, as the latter can be seamlessly combined with any DP fine-tuning pipelines. Our evaluation used the strong DP fine-tuning baseline from [1], and showed consistent results across models (GPT2 small, medium, large) and datasets (BookSum & Enron Email).
>
> In response to the suggestion, we reran our evaluation using the pipeline from the referenced ICLR'22 work on Enron Email. Consistent with our previous findings, random chunk works best for user-wise DP-SGD, and longest sequence selection for group privacy.  The performance of both mechanisms, using their respective best selection strategies, is shown below and aligns with our submission results.
>
> |ε|Group privacy|User-wise DP-SGD|
> |---|---|---|
> |1.0|33.48|33.21|
> |3.0|31.09|30.76|
> |8.0|28.34|28.07|
>
> ### Q3 Assuming user info available: might be missing in real applications
>
> **A**: We study the standard notion of user-level DP (Liu'20, Levy'21, Ghazi'24), which is increasingly important [2] as example-level DP alone is insufficient to provide the same level of protection for users contributing multiple examples. While our current evaluation uses user identifiers to group examples due to the lack of dedicated public datasets for such evaluation, in real world settings, these methods do not require user identifiers as long as there is a mechanism to group examples based on user ownership.
>
> [1] Yu et al. Differentially Private Fine-tuning of Language Models. ICLR’21
>
> [2] Case et al. A distributed attribution and aggregation protocol.

---

> > ### Author Response · Authors · 2024-06-03
> > **Follow-up**
> >
> > Dear Reviewer sVEv,
> >
> > Thank you once again for taking the time to share your suggestions on our work. As the discussion period is nearing its end, we want to ensure that we have adequately addressed all your comments. Specifically, do you still have concerns regarding:
> > - Q1: Our contribution, where we contribute a **comprehensive evaluation suite of user-level DP implementations for LLMs and well-motivated data selection strategies**.
> > - Q2: Performance when integrated with other DP fine-tuning methods, where we present **new results for our evaluation when incorporated into the DP fine-tuning pipeline you suggested**.
> > - Q3: The availability of user information in real applications, where we adhere to the **standard user-level DP notion** that can still function as long as there is a mechanism to group examples based on user ownership.
> >
> > We look forward to your feedback.
> >
> > Best regards,
> >
> > Authors of Submission933

---

> > > ### Comment · Reviewer_sVEv · 2024-06-04
> > > **Thanks for the response**
> > >
> > > Thanks for the informative response. I've changed my rating.

---

> > > > ### Author Response · Authors · 2024-06-04
> > > > **Thank you**
> > > >
> > > > Thank you for taking the time to read our response and for raising your score.
> > > >
> > > > If you have any further questions or need additional clarification, please feel free to reach out.

---

### Official Review · Reviewer_pxQP · 2024-05-11

**Rating:** 7
**Confidence:** 4
**Ethics Flag:** 1

**Summary:**

In this paper, the authors present a study on the application of user-level differential privacy (DP) in the context of fine-tuning large language models (LLMs) on natural language generation tasks.  It is a critical issue in the field of machine learning, specifically the privacy concerns associated with fine-tuning LLMs on sensitive data. The document is well-structured, with a clear abstract, introduction, methodology, experimental setup, results, and conclusions. Experimental results on Enron Email dataset and BookSum dataset were given to show the effectiveness of the proposed method.

**Reasons To Accept:**

1. The paper tackles the concept of user-level DP in the fine-tuning of LLMs, which is a departure from the traditional example-level DP.

2. The authors provide a systematic empirical evaluation of two user-level DP mechanisms.

**Reasons To Reject:**

1. The paper mentions that User-wise DP-SGD may require modifications to the data loading pipeline, which could be a barrier to adoption in practice.

2. The computational overhead introduced by the differential privacy mechanisms is mentioned but not thoroughly analyzed, particularly for larger models or datasets.

---

> ### Author Rebuttal · Authors · 2024-05-30
>
> ### Q1 Implementation of User-wise DP-SGD
> > The paper mentions that User-wise DP-SGD may require modifications to the data loading pipeline, which could be a barrier to adoption in practice.
>
> **A**: Thanks for the comment. Please note that an efficient implementation of user-wise DP-SGD can be achieved with only **minor changes** to the data loading pipeline. Specifically, it entails:
> - Modifying the per-example dataloader to a per-user dataloader (line 5 in Algorithm 2), which only requires implementing a customized dataloader with a few additional lines of code.
> - Fetching training examples for each user and taking the gradient average per user on the fly during each gradient step (lines 6~10 in Algorithm 2).
>
> Fortunately, since both steps can be implemented with minimal effort, user-wise DP-SGD can be easily adopted in practice.
>
> ### Q2 Computational overhead
> > The computational overhead introduced by the differential privacy mechanisms is mentioned but not thoroughly analyzed, particularly for larger models or datasets.
>
> **A**: Thanks for the comment. The main overhead of group privacy training methods is introduced by the per-example gradient clipping and noise addition (lines 11-13 in Algorithm 1). For the user-wise DP-SGD method, the primary overhead stems from the user-wise data sampling (line 5 in Algorithm 2), as well as per-user gradient clipping and noise addition (lines 8-9 in Algorithm 2).
>
> To provide a more quantitative analysis, we present the following table showcasing the training time (in seconds) per step for different models and training methods with $k=10$ on the Enron Email dataset. The results are obtained using LoRA fine-tuning with rank being 32 and a batch size of 1024, averaged across 100 training steps, on a single NVIDIA A100-80G GPU.
>
> | Model | Non-private training | Group privacy | User-wise DP-SGD |
> |:---:|:---:|:---:|:---:|
> | GPT2 | 11.95 | 12.70 (1.06x) | 12.82 (1.07x) |
> | GPT2-medium | 30.90 | 32.00 (1.04x) | 32.85 (1.06x) |
> | GPT2-large | 62.57 | 64.80 (1.04x) | 68.45 (1.09x) |
>
> As shown in the table, both methods introduce very little computational overhead (4%–9%) compared to non-private training for the GPT-2 model, regardless of the size.

---

> > ### Author Response · Authors · 2024-06-03
> > **Follow-up**
> >
> > Dear Reviewer pxQP,
> >
> > Thank you once again for taking the time to share your suggestions on our work. As the discussion period is nearing its end, we want to ensure that we have adequately addressed all your comments. Specifically, do you still have concerns regarding:
> > - Q1: The implementation of user-wise DP-SGD, where we clarify that it **can be implemented with minimal effort**.
> > - Q2: The computational overhead, where we report the training time for your reference and observe **minimal computational overhead** (4%–9%) compared to non-private training for the GPT-2 model of various sizes.
> >
> > We look forward to your feedback.
> >
> > Best regards,
> >
> > Authors of Submission933

---

### Official Review · Reviewer_mJeH · 2024-05-16

**Rating:** 6
**Confidence:** 3
**Ethics Flag:** 1

**Summary:**

In this work, the authors study language model fine-tuning with differential privacy guarantees. Since each user generally contributes multiple text sequences to the training corpus, to ensure equal privacy guarantees across users, the privacy unit needs to be set at the user level. The authors identify and systematically evaluate two mechanisms, Group Privacy and User-wise DP-SGD, for achieving user-level differential privacy when fine-tuning large language models on natural language generation tasks. For each mechanism, the authors investigate data selection strategies to improve the trade-off between privacy and utility.

**Questions To Authors:**

see weakness.

**Reasons To Accept:**

1. The authors present a systematic empirical evaluation of User-level DP for fine-tuning LLMs on natural language generation tasks.

2. The authors therefore study user-level DP motivated by applications where it necessary to ensure uniform privacy protection across users.

**Reasons To Reject:**

1. The technical novelty is limited. What is the challenge to adapt User-wise DP-SGD in LLM, there are many existing User-wise DP-SGD methods.

2. Lack of baselines. It is better to add some recent baseline to demonstrate the advantage of the proposed method.  For example: Large Language Models Can Be Strong Differentially Private Learners. 2022 ICLR

---

> ### Author Rebuttal · Authors · 2024-05-30
>
> ### Q1 Limited technical novelty
> > What is the challenge to adapt User-wise DP-SGD in LLM given there are many existing methods?
>
> **A**:  We outline three challenges in adapting existing methods and place our contributions in this context.
>
> 1. **Lack of a systematic evaluation**: Prior work (McMahan'18; Kairouz'21; Xu'23) has neither systematically studied user-level DP implementation choices nor their performance in LLMs. Many applications used naive group privacy (Amin'19; Levy'21) which leads to worse utility for smaller ε's. (see Sec.6).
>
> - Our contribution: We systematically evaluated the two mainstream mechanisms for user-level DP in LLMs. We also empirically show the overly restrictive assumptions of Asi'23, an algorithm with a theoretically optimal excess rate.
>
> 2. **Lack of reliable testbeds** for user-level DP methods in LLM applications for effective comparison
>
> - Our contribution: We derived two testbed datasets, Enron Email and BookSum, from existing non-DP-purpose datasets to simulate real-world use cases and benefit future research.
>
> 3. **Overlooked data selection strategy** when adapting user-level DP to LLMs
>
> - Our contribution: We proposed and evaluated data selection strategies for both group privacy and user-wise DP-SGD, showing significant performance gap based on the strategy used - e.g., on Enron Email w/ ε=1.0, perplexity ranges from 34.01 for the best strategy to 35.68 for the worst (Table 4a).
>
> ###  Q2 Lack of baselines
> >e.g. Large Language Models Can Be Strong Differentially Private Learners. ICLR’22
>
> **A**: Please note that the choice of DP fine-tuning recipe is **orthogonal** to the user-level DP algorithms, as the latter can be seamlessly combined with any DP fine-tuning pipelines. Our evaluation used the strong DP fine-tuning baseline from [1], and showed consistent results across models (GPT2 small, medium, large) and datasets (BookSum & Enron Email).
>
> In response to the suggestion, we reran our evaluation using the pipeline from the referenced ICLR'22 work on Enron Email. Consistent with our previous findings, random chunk works best for user-wise DP-SGD, and longest sequence selection for group privacy.  The performance of both mechanisms, using their respective best data selection strategies, is shown below and aligns with our submission results.
>
> |ε|Group privacy|User-wise DP-SGD|
> |-|-|-|
> |1.0|33.48|33.21|
> |3.0|31.09|30.76|
> |8.0|28.34|28.07|
>
> [1] Yu, et al. Differentially Private Fine-tuning of Language Models. ICLR’21

---

> > ### Comment · Reviewer_mJeH · 2024-05-31
> >
> > Thanks for the response. It addressed most of my concerns. I will raise my score.

---

> > > ### Author Response · Authors · 2024-05-31
> > > **Thank you for your response**
> > >
> > > Thank you for taking the time to read our response and for raising your score. We're glad to hear that our response addressed most of your concerns.
> > >
> > > If you have any further questions or need additional clarification, please feel free to reach out.

---

### Decision · Program_Chairs · 2024-07-10

**Decision:**

Accept

**Comment:**

Existing evaluations of privacy-preserving LLMs often treat each example (text sequence) as the privacy unit, leading to uneven privacy guarantees when contributions per user vary. To address this, the authors study user-level differential privacy (DP), driven by the need for uniform privacy protection across users. They conduct a systematic evaluation of user-level DP for LLM fine-tuning on natural language generation tasks under two mechanisms for achieving user-level DP guarantees—Group Privacy and User-wise DP-SGD. They investigate design choices such as data selection strategies and parameter tuning to optimize the privacy-utility tradeoff. Following the author response and author-reviewer discussions, this paper has received unanimous support from the reviewers. Therefore, I recommend acceptance.